# In Vitro Modeling of Diurnal Changes in Bone Metabolism

**DOI:** 10.3390/ijms26167699

**Published:** 2025-08-08

**Authors:** Sabrina Ehnert, Xiang Gao, Maximilian Heßlinger, Niklas R. Braun, Kevin A. Schulz, Denise Jahn, Fabian Springer, Andreas K. Nussler

**Affiliations:** 1Siegfried Weller Research Institute, BG Unfallklinik Tuebingen, Department of Trauma and Reconstructive Surgery, University of Tuebingen, Schnarrenbergstr. 95, D-72076 Tuebingen, Germany; sabrina.ehnert@med.uni-tuebingen.de (S.E.); xianggao23@gmail.com (X.G.); maximilian.hesslinger@gmail.com (M.H.); nbraun2@bgu-tuebingen.de (N.R.B.); kschulz@bgu-tuebingen.de (K.A.S.); 2Julius Wolff Institute, Berlin Institute of Health at Charité-Universitätsmedizin Berlin, D-13353 Berlin, Germany; denise.jahn@charite.de; 3Department of Diagnostic and Interventional Radiology, University of Tuebingen, Hoppe-Seyler-Str. 3, D-72076 Tuebingen, Germany; fabian.springer@med.uni-tuebingen.de

**Keywords:** osteoblast, osteoclast, in vitro model, circadian rhythm, BMAL1, CLOCK

## Abstract

There is evidence that bone health is closely linked to a functioning circadian rhythm. Most of the evidence comes from mice, which may exhibit some species-specific differences from humans due to their nocturnal lifestyle. To address the current lack of human model systems, the present study aimed to develop an in vitro model system that can represent diurnal changes in bone metabolism. The model is based on co-cultured SCP-1 and THP-1 cells that serve as osteoblast and osteoclast precursors, respectively. Diurnal effects were induced by replacing the FCS in the differentiation medium with human serum pools (HSPs) obtained in the morning, noon, or evening. The model system was tested for cell viability, gene expression, and osteoblast and osteoclast function. The replacement of the FCS with the HSPs increased viability and induced expression changes in circadian clock genes in the model system. Resulting alterations in osteoblast and osteoclast function led to a gradual increase in mineral density and stiffness when 3D co-cultures were differentiated in the presence of the HSPs collected in the morning, noon, or evening, respectively. Here, we present for the first time an in vitro model that can present diurnal changes in bone metabolism in the form of a snapshot. With the simple use of HSPs, this model can be used as a platform technique to investigate bone function in various situations, taking into account the time of day.

## 1. Introduction

The term “circadian rhythm” refers to a natural oscillatory process in the human body that operates on a 24 h cycle, influenced by changes in external light intensity throughout the day. The circadian clock system comprises the central clock, located in the suprachiasmatic nucleus (SCN), and peripheral oscillators. The SCN, located in the hypothalamus in the brain, receives input from the eyes about fluctuations in light intensity throughout the day [1]. Depending on the stimulus, the SCN controls direct behavioral (e.g., appetite or tiredness), autonomic (e.g., blood pressure or kidney function), and neuroendocrine (hormones, e.g., melatonin, insulin or growth hormone) circadian rhythms. Although there is ongoing debate regarding the specific substances in the serum that are responsible for synchronizing the peripheral circadian rhythms, it is widely accepted that their serum concentrations vary depending on the time of day, highlighting the role of serum in the synchronization process [2]. The resulting systemic signals synchronize molecular clocks in peripheral tissues and thus regulate their physiological rhythms crucial for health [1,3], e.g., bone formation and resorption.

The molecular clock is regulated by genes of the circadian rhythm core loop, including *BMAL1*, *CLOCK*, *PER1*–*3*, and *CRY1*–*2*. BMAL1 binds together with CLOCK to E-box domains on gene promoters such as those contained in *PER1*–*3* and *CRY1*–*2* and induces the expression of these genes. Upon translation, dimers of PERs and CRYs are phosphorylated by casein kinases, enabling their translocation to the nucleus, where they inhibit CLOCK:BMAL1-mediated transcription and finally their own expression (feedback loop) [4]. The stability of PERs and CRYs is regulated by ubiquitin-dependent proteasomal degradation. Additional transcription–translation negative-feedback loops (TTFLs) and the complementary action of several proteins stabilize this rhythm. NPAS2, for example, is a paralog of CLOCK and can substitute its function under several circumstances and might adapt the circadian clock on metabolic processes [5]. Another feedback loop is mediated by the nuclear receptors REV-ERBα and -β (regulated via CLOCK/BMAL1), which compete with the retinoic acid-related orphan receptors, RORα, -β, and -γ for binding to RRE-elements on the *BMAL1* gene promoter, regulating BMAL1 transcription.

Key hormones that regulate bone metabolism, such as melatonin, insulin, growth hormone, and parathyroid hormone, exhibit diurnal fluctuations. The same holds for calcium, phosphate, or magnesium levels in the blood, highlighting the close coupling between the circadian system and skeletal homeostasis [6,7]. Therefore, it is not surprising that over a quarter of all transcripts in bone tissue have been reported to show circadian rhythmicity, including genes for osteoblastogenesis and osteoclastogenesis, osteogenic cytokines, and signaling proteins (for review, see [6]). For example, in human serum, the osteoblast markers Osteocalcin and PICP (carboxy-terminal propeptide of type I procollagen) have been reported to reach their peak levels during the night and slightly earlier than the collagen type I degradation products CTX (collagen type I C-telopeptide) and NTX (N-terminal cross-linked telopeptide of type I collagen), which are characteristic for osteoclast function, which has been reported to reach their peak levels overnight or early in the morning [6,8]. The amplitude of the CTX and NTX rhythms may be lowered with fasting or with anti-resorptive therapy, but is generally unrelated to sex, age, or physical activity [9]. Overall, circadian changes in osteoclast-related markers were reported to be more pronounced than that of osteoblast-related markers [8], suggesting osteoclasts are a critical target. However, a murine model with cell specific conditional knockdown of BMAL1 in osteoclasts or mesenchymal stromal cells (MSCs) proposes that osteoblasts might be more prone to alterations in circadian rhythm than osteoclasts [10]. Disruption of the circadian rhythms, e.g., through shift work or sleep deprivation in humans, or genetic knockout in mice, has been linked to abnormal bone metabolism, osteoporosis, and increased fracture risk [6,7,11]. A significant part of research on this topic has employed mouse models [12,13], as these animals can be exposed to defined light–dark cycles to induce or modulate circadian rhythms. However, it must be taken into account that mice are nocturnal, while humans are diurnal, which affects the sleeping and eating times involved in the regulation of the circadian rhythm. In contrast, taking tissue samples from humans is much more difficult than from mice and can often only be achieved once from diseased individuals, due to ethical restrictions. Therefore, indirect methods, such as the analysis of factors circulating in the blood (e.g., bone turnover markers) within a 24 h cycle, have often been used in human studies [8].

The use of in vitro models is critically discussed among researchers in the field, as cultured cells gradually lose their synchronicity due to the absence of rhythmic stimulation from the SCN and its downstream signaling cues. Murine ex vivo bone cultures display a gradually decline of rhythmic gene expression during culture time that can be restored by external factors like PTH supplementation or media change with B-27 supplement, even after several weeks of cultivation, suggesting persistent endogenous mechanisms that can be reactivated upon stimulation [14,15]. For bone as tissue, various mono- and co-culture models exist that may be used to display human bone metabolism [16], with a clear advantage of co-cultures that take into consideration the interplay between osteoblasts/osteocytes (bone formation) and osteoclasts (bone resorption). However, these co-culture models are usually composed of human cell lines that are either immortalized or cancer-derived and therefore often exhibit aberrant circadian clock functions [17]. Therefore, the present study aimed at establishing culture conditions that can reflect diurnal changes in bone metabolism (bone formation and resorption), characteristic for a specific time of day.

## 2. Results

### 2.1. Lack of Rhythmicity in Expression of Circadian Clock Genes in Conventional Osteoblast-Osteoclast Co-Culture

Classically, the 2D in vitro bone model comprised THP-1 cells (myeloid osteoclast precursors) and SCP-1 cells (osteoblast precursors) cultured in the presence of 2% fetal calf serum (FCS—medium routinely replaced Mondays, Wednesdays, and Fridays in the morning). Compared to the growth medium, the differentiation medium had a reduced FCS content in order to synchronize the cells and to favor bone cell differentiation. At day 21 of co-culture, total RNA was collected in the morning (7–8 am), noon (1–2 pm), and evening (7–8 pm) to examine the expression of genes involved in the circadian rhythm core loop: *BMAL1*, *CLOCK*, *NPAS2*, and their negative feedback regulators *CRY1*, *PER1* and *PER2* (Figure 1). Overall, no significant differences in gene expression were observed between the groups, except for the *CLOCK* gene, whose expression was lowest when mRNA was collected in the evening (Figure 1B).

### 2.2. Adaptation of the Bone Cell Co-Culture

In order to introduce diurnal changes in the bone cell co-culture model, it was tested whether the fetal calf serum (FCS) in the differentiation medium can be replaced with a human serum pool (HSP) from N = 10 donors—a schematic overview of the experimental setup is given in Figure 2A. The 10 donors were healthy volunteers that donated their blood within one working day with low to moderate physical exertion. Blood was obtained in the morning (7–8 am) before breakfast, at noon (1–2 pm) at least 1 h after lunch, and in the evening (7–8 pm) before supper. The donor group was composed of six males (60%) and four females (40%). The donors had a mean age of 28.6 ± 5.9 years (21–43 years) and a mean body mass index (BMI) of 24.2 ± 3.7 kg/m^2^ (18.2–29.2 kg/m^2^). None of the donors took daily medication.

HSP concentrations of 1%, 2%, 3%, 4%, and 5% were tested. HSP concentrations above 2% resulted in an overgrowth of the cells as soon as day 10 of culture, such that functional assays could not be performed. Differentiation in the presence of only 1% and 2% HSP resulted in significantly higher mitochondrial activity (resazurin conversion) and total protein content (sulforhodamine B (SRB) staining) than the classical differentiation in the presence of 2% FCS (Figure 2B,C). As an early osteoblast marker, alkaline phosphatase (ALP) activity was detected on day 14 of culture. When normalized to the total protein content, the presence of 2% HSP decreased the ALP activity compared to with the conventional differentiation protocol (Figure 2D). A similar trend was observed for the activity of the two osteoclastic markers carbonic anhydrase II (CAII—Figure 2E) and tartrate-resistant acidic phosphatase (TRAP5b—Figure 2F). There was no significant difference in ALP, CAII, and TRAP5b activity when comparing the differentiation with 2% FCS and 1% HSP—which qualified this HSP concentration for the following model testing.

### 2.3. HSP Obtained in the Evening (7–8 Pm) Induced Cell Growth Stronger than That with HSP Obtained in the Morning (7–8 Am) or Noon (1–2 Pm)

THP-1 and SCP-1 cells were co-cultured in osteogenic differentiation medium containing 1% human serum pools obtained from 10 identical donors in the morning (7–8 am), noon (1–2 pm), or evening (7–8 pm) of the same day. Viability of the co-culture was determined by mitochondrial activity and total protein content on days 7, 14, and 21 of culture. Overall, the increase in cell number was highest in the evening group, with significant difference compared to the noon group (Figure 3A,B). Additionally, on day 21 of co-culture, the expressions of genes related to cell proliferation and growth (*MKI67*, *TPX2*, and *TOP2A*) were analyzed. In line with the results from resazurin conversion and SRB staining, the expression of these three genes was highest in the evening group, with significant difference to the noon group for *TPX2* and *TOP2A* (Figure 3C–E).

### 2.4. Highest Osteoblast Function Observed in Co-Cultures Differentiated in the Presence of HSP Obtained in the Evening (7–8 Pm)

The same co-cultures were then analyzed for osteoblast function. On day 21 of co-culture, the expression of the two osteogenic key transcription factors *RUNX2* and *SP7* (encoding Osterix) was quantified by qRT-PCR. While the expression of the early osteoblast differentiation marker *RUNX2* was highest in the morning group (with significant differences to the noon and evening group), the inverse was observed for the late osteoblast differentiation marker *SP7*, suggesting faster osteogenic differentiation in the noon and evening group (Figure 4A,B). In line with the qRT-PCR results, ALP activity showed a faster increase in the noon and evening group, which was significant over time when compared to the morning group (Figure 4C). Most of the mineralized matrix was formed in the evening group, as visualized (Figure 4D) and quantified by von Kossa (Figure 4E) and Alizarin Red (Figure 4F) staining, with significant difference to the morning group.

### 2.5. Composition of M-CSF, RANKL, and OPG in the HSP Favors Osteoclastogenesis in Co-Cultures Differentiated in the Presence of HSP Obtained in the Morning (7–8 Am)

M-CSF, RANKL and OPG are key regulators of osteoclastogenesis that may show circadian variation in the serum. Therefore, their concentration was determined using a dot blot analysis with the three HSPs used for differentiation (Figure 5A). The quantification of the dot blot signals showed minimal, insignificant variation in the serum levels of M-CSF, RANKL, and OPG. The overall trend showed the highest serum levels of M-CSF and RANKL in the morning group. Inversely, OPG serum levels tend to increase with the time of day (Figure 5B). Taken together, this indicates that the presence of HSP obtained in the morning will favor osteoclastogenesis compared to HSP obtained at noon or in the evening. To investigate osteoclastogenesis, the expression of the key osteoclastic transcription factor *NFATc1* was quantified by qRT-PCR. The expression of *NFATc1* was highest in the morning group, with significant difference to the evening group (Figure 5C). In line with this, CAII and TRAP5b function were significantly higher in the morning group when compared to the noon or evening group (Figure 5D,E).

### 2.6. Three-Dimensional Co-Culture Showed Highest Bone Mineral Density and Stiffness When Differentiated in the Presence of HSP Obtained in the Evening (7–8 Pm)

To better investigate the interplay between osteoblasts and osteoclasts, THP-1 and SCP-1 cells were co-cultured in 3D using cryogel scaffolds. After 21 days of co-culture, the mineral density of the cryogel carriers was determined by computer tomography (CT)—Figure 6A shows representative 3D reconstructions of the core of the cryogels. Densitometric analysis of the CT images revealed significantly higher mineral density in the evening group compared to the morning group or noon group (Figure 6B). Further, the stiffness of the cryogels was determined using a ZwickiLine Z 2.5TN material testing machine. After 21 days of co-culture, the cryogels in the morning group were significantly less stiff compared to the cryogels of the noon group or evening group (Figure 6C).

### 2.7. Replacement of FCS with HSP in the Osteogenic Differentiation Medium Affects the Expression of the Genes Involved in the Circadian Rhythm Core Loop

As described before, the expression of genes involved in the circadian rhythm core loop—*BMAL1*, *CLOCK*, *NPAS2*, and their negative feedback regulators *CRY1*, *PER1*, and *PER2*—was determined after 21 days of co-culture in the presence of HSPs obtained in the morning (7–8 am), noon (1–2 pm), or evening (7–8 pm). In contrast to the conventional co-culture, replacing the 2% FCS in the differentiation medium with 1% HSP resulted in an altered expression of the six genes. The expression of *BMAL1* (Figure 7A), *CLOCK* (Figure 7B), and *NPAS2* (Figure 7C) was highest in THP-1 cells and SCP-1 cells co-cultured in the presence of HSP obtained at noon, with significant difference to the co-cultures from the evening group. The expression of *CRY1* was highest in the morning group, with significant difference to the evening group (Figure 7D). The inverse was observed for the expression of *PER1* (Figure 7E) and *PER2* (Figure 7F), which showed the highest expression in the evening group.

## 3. Discussion

In the past years, there is more and more evidence that bone health underlies a controlled circadian rhythm [6,7,11]. However, due to limitations in tissue sampling in humans, most of the knowledge about mechanistic regulations in bone circadian rhythm is obtained from mouse models [12,13]. However, mice are nocturnal, with significantly altered sleeping and eating times compared to diurnal humans. As these two factors may critically influence circadian rhythm, findings from mouse models may not directly translate to the human situation. Therefore, human in vitro models could be a suitable alternative for mechanistic analyses, especially as murine ex vivo bone cultures and isolated bone cells have suggested that bone metabolism exhibits an endogenous rhythm that can be synchronized with external cues [14,18]. However, to our knowledge, so far no human in vitro bone cell culture model has been able to display the diurnal differences in bone formation and resorption expected from measurements of bone turnover markers in human blood [8]. The model used in this study is based on a recently published co-culture of osteogenic SCP-1 cells and osteoclasts derived from THP-1 cells [16], which enables the parallel investigation of both bone formation and resorption. Culturing the cells with the published classical differentiation protocol failed to display diurnal changes in the expression of the genes controlling the circadian rhythm core loop (*BMAL1*, *NPAS2*, *PERs*, and *CRYs*). Only the expression of *CLOCK* displayed significant differences between the three chosen time points (morning, noon, evening), with the lowest levels in the evening group. This is in line with other studies showing that in cultivated murine bone cells or tissues, cells lose their rhythmic gene expression with culture time and need additional stimulation to be synchronized [14,15,18]

Considering the role of serum in linking central and peripheral circadian rhythms within the human body [1,19], we chose to replace the FCS in the differentiation medium with HSPs, obtained from the same 10 healthy individuals in the morning (7–8 am), noon (1–2 pm), and evening (7–8 pm) within one day, to diminish confounding factors, e.g., hormonal changes in female donors. Replacing FCS with HSPs increased overall cell viability and proliferation, so that the addition of 1% HSP in the differentiation medium was sufficient for the co-culture to mature with measurable osteoblast and osteoclast function. Further, the diurnal changes in HSP composition affected both the cell proliferation and differentiation.

The replacement of the 2% FCS with 1% HSP in the differentiation medium was also sufficient to induce the expression of some of the genes controlling the circadian rhythm core loop. The expression of *BMAL1*, *CLOCK*, and *NPAS2* was highest in the co-cultures differentiated in the presence of the HSP obtained at noon, with significant decline towards the co-cultures differentiated in the presence of the HSP obtained in the evening. This is in line with observations from in vivo bone samples, where *BMAL1* and *CLOCK* expression increases with the duration and intensity of the mice’s light exposure. The expression of *CRY1*, *PER1*, and *PER2* was inversely regulated in the same samples [20]. In our model, this is only true for the expression of *PER1* and *PER2*. *CRY1* expression was shifted from that of the two *PERs*, with the highest expression observed in co-cultures differentiated in the presence of the HSP obtained in the morning. In skeletal muscle (C2C12) cells, the expression of these genes was similarly counter-regulated as in our model. Furthermore, their expression levels were affected by mechanical stimulation, i.e., reduction in *PERs* and *CRYs* and induction of *BMAL1* and *CLOCK* [21], suggesting that physical activation, usually during the day in humans, is an independent stimulus. The changes in clock gene expression also correlated with the cell proliferation [21]. A similar correlation is seen in our cells only when they are differentiated in the presence of the HSP obtained at noon. Inversely, knockdown or BMAL1 induced cell senescence in primary bone-marrow-derived MSCs [22]. Proliferation was strongest in the co-cultures differentiated in the presence of the HSP collected in the evening, as observed by increased mitochondrial activity and total protein contents and supported by an elevated expression of *MKI67*, *TOP2A*, and *TPX2*, involved in proliferation [23]. In the investigated model, proliferation can be mostly linked with the osteogenic SCP-1 cells, as THP-1 cells lose most of their ability to proliferate after activation with PMA (phorbol 12-myristate 13-acetate) [24]. In the human body, glucose is the major source of energy for bone development, growth, and maintenance [25]. Since mean blood glucose levels normally increase throughout the day with every meal consumed [26], this could explain why the co-cultures in the evening group showed highest cell proliferation, especially as our healthy volunteers donating the HSP had no breakfast before the first blood sampling (fasting blood glucose levels). Furthermore, glucose uptake via GLUT1 has been reported to synergistically interact with RUNX2 to orchestrate osteoblast differentiation and bone formation [27]. This hypothesis supports our data showing faster osteogenic differentiation in the co-cultures of the evening group. While co-cultures in the morning group still expressed considerable amounts of the early osteogenic transcription-factor *RUNX2* after 21 days of culture, its expression in the co-cultures of the noon and evening groups was already replaced with its downstream target *SP7*, which encodes the late osteogenic transcription factor osterix [28]. Regarding the functional site, these results are supported by an early increase in ALP activity and associated formation of mineralized matrix in co-cultures of the noon group and especially the evening group. In primary murine bone-marrow-derived MSCs, the expressions of *RUNX2*, *SP7*, and osteogenic differentiation were dependent on functional BMAL1 [22], which was shown to positively affect osteogenic differentiation of MC3T3-E1 cells in vitro [29].

In vivo male mice with postnatal osteoblast-specific deletion of BMAL1 under the control of the *COL1A1*-promotor showed multiple abnormalities in bone metabolism, resulting in a progressive decrease in cortical bone mass and a concomitant unexpected increase in trabecular bone mass. With increasing age, these mice developed kyphoscoliosis and malformed intervertebral joint disease [30]. In contrast, when BMAL1-knockout was induced in MSCs of the hind limb under the control of the *PRX1*-promotor, both cortical and trabecular bone mass were significantly decreased [10]. Interestingly, osteoclast-specific deletion of BMAL1 under the control of the *CTSK*-promotor did not affect trabecular or cortical bone mass [10], suggesting cell-specific circadian effects. In line with this, it has been noted that skeletal aging in male mice is accompanied by a decline in *BMAL1*, especially in bone marrow endothelial cells. Deletion of BMAL1 in CDH5-positive bone marrow endothelial cells resulted in massive decline in trabecular bone mass [31]. The authors proposed an excessive breakdown of FBN1 as a possible underlying mechanism. The resulting persistent activation of TGF-β/SMAD3 signaling may exhaust MSCs while activating osteoclasts. Unlike the described detrimental effects of BMAL1-deletion on bone, mice deficient in PERs and CRYs usually display increased bone mass [32], supporting the negative feedback loop controlling the expression of clock genes. The authors suggested that the expression of *PERs* and *CRYs* is regulated by leptin, which has previously been shown to inhibit bone formation via a hypothalamic relay function [33]. Circadian timing in peripheral osteoclasts was reported to be mediated by glucocorticoids, directly inducing the expression of *PERs* and *NFATc1* [34]. In humans, glucocorticoid (cortisol) levels follow a diurnal rhythm, with peak serum levels early in the morning, which could explain the increased expression of *NFATc1* and elevated osteoclast activity in our co-cultures differentiated in the presence of HSP collected in the morning.

Osteoclast formation is further controlled by M-CSF, RANKL and its inhibitor OPG. The HSPs used in our co-culture showed only small insignificant differences in M-CSF, RANKL, and OPG levels, which is in line with other studies investigating circadian alterations in bone markers in human blood—for review, see [8]. The overall change in the composition of OPG (highest in the evening), RANKL, and M-CSF (both highest in the morning) in the three HSPs used in this study suggests increased osteoclast formation in the morning group. This was confirmed by the increased *NFATc1* expression as well as increased CAII and TRAP5b activity in co-cultures differentiated in the presence of the HSP obtained in the morning. However, M-CSF, RANKL, and OPG are not only provided by the HSP in our model but can also be produced by the SCP-1 cells [35]. While there is no report in the literature on BMAL1 effects on *M-CSF* or *RANKL* expression, *OPG* expression was reported to be directly controlled by BMAL1 in osteoblastic cells [36], supporting our finding that OPG levels show an increasing trend throughout the day. Further, it has been reported that melatonin in a BMAL1-dependent manner induces apoptosis in RAW264.7 cells, mouse mononuclear macrophage leukemia cells known to serve as osteoclast precursors. In this way, melatonin treatment could improve osteoporosis in ovariectomized mice [37]. These studies support our findings showing a steady increase in mineral density and stiffness when 3D co-cultures were differentiated in the presence of the HSPs collected in the morning, noon, or evening, respectively. This effect might be even more pronounced when our model system is differentiated in the presence of HSP obtained at night, when melatonin levels reach their maximum. Given that the peak serum concentration of bone formation markers was described a little bit earlier for the night group compared to the peaks of bone resorption markers, our data supports the hypothesis that serum components might be responsible for this modulation. We showed that serum taken in the evening is more supportive for osteoblast activity, maybe reflecting the start of the activation of these cells and morning serum supports osteoclast activity, which might be related to residual stimulating effects on bone resorption. More detailed analyses of sera from different times of the night might be helpful to identify the responsible mediators and to characterize the timed action of osteoblasts and osteoclast during the night.

Taken together, we here present for the first time an in vitro model that can mirror diurnal changes in bone metabolism in vivo. However, it needs to be considered that our model provides a snapshot of the diurnal changes and no rhythmic alterations. Considering this potential limitation, the model can be used as a platform to investigate diurnal bone function in various situations, such as in investigating the influence of comorbidities (e.g., chronic obstructive pulmonary disease (COPD), psychiatric or neurodegenerative disorders), lifestyle habits (e.g., shift work, jet lag, or drug abuse), or medications (e.g., melatonin, benzodiazepines, modafinil, or methylphenidate) with a simple setup of comparing different age- and gender-matched groups. However, knowing the diurnal alterations in osteoblast and osteoclast activity, the blood sampling should be narrowly timed with early sampling in the morning for studies focusing on osteoclast regulation and sampling late in the evening for studies focusing on osteoblast regulation. Further, care should be taken in the timing of food intake, as the resulting alterations in blood glucose levels might affect the proliferation and differentiation of the cells in the model system [38]. While such a group comparison can be carefully planned to reduce the limitations of the model system, the situation is more complex when the model should be used for analyzing intervention strategies with the same individuals at different time points.

## 4. Materials and Methods

### 4.1. Human Serum Pool

Blood sampling was conducted in accordance with the Declaration of Helsinki and approved by the Ethics Committee of the University of Tübingen, Number 844/2020B02. A total of 10 healthy young volunteers participated in the study, with blood samples collected from each participant in the morning (7–8 am), noon (1–2 pm), and evening (7–8 pm). The blood was incubated at room temperature for 30 min to facilitate blood clotting and then centrifuged at 2000× *g* for 10 min (room temperature). The obtained sera were pooled using equal volumes and then stored at −80 °C in aliquots to prevent repetitive freeze and thaw cycles.

### 4.2. Osteoblast–Osteoclast Co-Culture

THP-1 cells (ACC16, DSMZ, Braunschweig, Germany) were expanded in RPMI 1640 Medium (R8758, Merck, Darmstadt, Germany), and 5% FCS and SCP-1 cells (provided by Prof. Dr. Matthias Schieker [39]) were expanded in Minimum Essential Medium Eagle Alpha (MEM α; AL081A, Omni Life Science, Bremen, Germany) with 5% FCS. Both cell lines were kept at 37 °C under a humidified atmosphere with 5% CO_2_. The medium was renewed twice a week. The generation of co-cultures followed the protocol described in [35].

Briefly, for the 2D co-cultures, 7.2 × 10^4^ THP-1 cells were seeded per cm^2^ in 300 µL THP-1 growth medium containing 200 nM Phorbol 12-myristate 13-acetate (PMA, Cay10008014-1, Biomol, Hamburg, Germany). After 24 h, non-attached THP-1 ells were removed, and SCP-1 cells (0.9 × 10^4^ cells/cm^2^) were seeded into the same wells with bone differentiation medium (300 µL/cm^2^ in 300 µL), with an even mixture of RPMI and MEM α, 2% FCS, or 1% human serum pool, as well as 200 µM L-ascorbic acid 2-phosphate (A8960-5G, Sigma, Saint Louis, MO, USA), 5 mM β-glycerophosphate (G9422-10, Sigma), 25 mM HEPES (HN78.2, Carl Roth, Karlsruhe, Germany), 1.5 mM CaCl_2_, and 5 µM cholecalciferol (95230, Sigma)). The cells were kept at 37 °C under a humidified atmosphere with 5% CO_2_ with medium renewal thrice a week.

For the 3D co-cultures, Platelet-Rich Plasma (PRP) Cryogel Scaffolds were prepared and sterilized following the published protocol [40]. Due to the increased surface of the scaffolds, cell numbers and medium volume were adapted to 8 × 10^4^ THP-1 cells and 1 × 10^4^ SCP-1 cells per scaffold and 500 µL, respectively. The plating procedure is described in detail in [35,41].

### 4.3. Quantitative Real-Time Polymerase Chain Reaction (qRT-PCR)

Total RNA was isolated from the osteoblast–osteoclast co-cultures (20 cm^2^ area per condition) using the phenol-chloroform extraction method with overnight incubation in isopropanol to increase the yield. After photometric quantification, RNA integrity was checked by agarose gel electrophoresis. A total of 500 ng of the total mRNA was converted into cDNA using the first-strand cDNA synthesis kit (K1612, ThermoFisher Scientific, Karlsruhe, Germany).

The qRT-PCRs were performed in a StepOnePlus^TM^ qPCR cycler using the Green Master Mix (2X) High Rox (#M3052.0500, Genaxxon, Ulm, Germany), with the primers and qPCR conditions listed in Table 1. B2M was used as housekeeping gene as it is intended to be the most stable in this setup based on the delta Ct (ΔCt), BestKeeper, GeNorm, and NormFinder methods [42]. Gene expression changes were determined using the ΔΔCt method and are displayed as z-scores.

### 4.4. Resazurin Conversion Assay

After 7, 14, and 21 days of 2D co-culture in 96-well-plates, mitochondrial activity was detected by resazurin conversion assay. Briefly, after being washed once with PBS, cells were covered with resazurin working solution (0.0025% *w/v* resazurin in PBS; 100 µL/well). After 60 min, resulting fluorescence (λ_EX_ 544 nm/λ_EM_ 590–10 nm) was detected with an Omega plate reader (BMG Labtech, Ortenberg, Germany) [16].

### 4.5. Sulforhodamine B (SRB) Staining

After 7, 14, and 21 days of 2D co-culture in 96-well-plates, total protein content was quantified by SRB staining. Briefly, cells were fixed with 100% ethanol (50 μL/well) for at least 1 h at −20 °C. After washing with tap water, cells were incubated with SRB solution (0.4% *w/v* resazurin in 1% acetic acid; 50 μL/well) for 30 min protected from light. Unbound SRB was removed by washing with 1% acetic acid. Bound SRB was resolved by adding 10 mM unbuffered TRIS solution (100 μL/well) and absorbance was measured immediately at λ 565 nm with background correction at λ 690 nm with an Omega plate reader [43].

### 4.6. Alkaline Phosphatase (AP) Activity

After 7, 14, and 21 days of 2D co-culture in 96-well-plates, cells were washed once with PBS before being covered with AP substrate solution (3.5 mM 4-nitrophenyl-phosphate, 50 mM glycine, 1 mM MgCl_2_, and 100 mM TRIS; pH = 10.5; 100 μL/well). After incubation for 2 h at 37 °C, absorbance was measured at λ 405 nm with an Omega plate reader [38].

### 4.7. Von Kossa Staining

After 21 days of 2D co-culture in 96-well-plates, formation of mineralized matrix was visualized by von Kossa staining. Briefly, cells were fixed with 100% ethanol (50 μL/well) for at least 1 h at −20 °C. After 3 washes with tap water, cells were covered with 3% silver nitrate solution (50 μL/well) for 30 min at ambient temperature. After another 3 washes with tap water, 50 μL sodium carbonate–formaldehyde solution (0.5 M sodium carbonate, 10% formaldehyde) were added per well for color development. Microscopic images were automatically obtained from 5 identical positions per well using the CelenaX microscope (logos biosystems, Anyang-si, Republic of Korea). Image analysis was performed using the ImageJ 1.47v software.

### 4.8. Alizarin Red Staining

After 21 days of 2D co-culture in 96-well-plates, the formation of mineralized matrix was quantified by Alizarin Red staining. Briefly, cells were fixed with 100% ethanol (50 μL/well) for at least 1 h at −20 °C. After washing with tap water, cells were incubated with staining solution (0.5% Alizarin Red in distilled water; pH = 4.0 50 μL/well) at room temperature for 30 min. Excess staining solution was removed by washing with tap water. Bound Alizarin Red was resolved with a 10% cetylpyridinium chloride solution, and absorbance was measured at λ 560 nm with an Omega plate reader [38].

### 4.9. Dot Blot Analysis

Specific proteins in the HSPs were detected by dot blot analysis. Briefly, 75 µL of a 1:10 dilution of the HSPs was applied on a wet nitrocellulose membrane with the help of a dot blotter (Carl Roth). After blocking membranes with 5% BSA for 1 h, primary antibody incubation (M-CSF #500-P44: Peprotech, Heidelberg, GER/RANKL sc-377079 and OPG sc-11383; both obtained from Santa Cruz Biotechnology, Heidelberg, Germany) was performed at 4 °C overnight. After incubation with the corresponding biotin-labeled secondary antibodies (Santa Cruz Biotechnology) for 2 h, membranes were incubated for 30 min with a 0.1% avidin-peroxidase solution. Chemiluminescent signals were detected by a CCD camera (INTAS, Göttingen, Germany) and quantified using the ImageJ 1.47v software [35,40].

### 4.10. Carbonic Anhydrase II (CAII) Activity

On days 7, 14, and 21 of 2D co-culture in 96-well-plates, cells were washed once with PBS before being covered with fresh CAII reaction buffer (12.5 mM TRIS, 75 mM NaCl, 200 mM 4-nitrophenyl acetate; pH = 7.5; 100 μL/well). Plates were immediately transferred to the Omega plate reader at 37 °C to quantify the absorbance change (λ 405 nm) over the following 15 min [38].

### 4.11. Tartrate-Resistant Acidic Phosphatase (TRAP) Activity Assay

TRAP activity was quantified in the collected culture supernatants of 2D co-cultures in 96-well plates on days 7, 14, and 21. In total, 30 μL of culture supernatant was mixed with 90 µL of TRAP substrate buffer (5 mM 4-nitrophenyl phosphate, 100 mM sodium acetate, 50 mM sodium tartrate; pH = 5.5). After incubation at 37 °C for 6 h, the reaction was stopped by adding 90 μL of 1 M NaOH. Absorbance was measured at λ 405 nm with an Omega plate reader [38].

### 4.12. Mineral Content of the PRP Scaffolds

After 21 days in co-culture, the mineral content of the scaffolds was evaluated through quantitative computed tomography (CT) scans using a clinical 128-slice CT scanner (SOMATOM Definition Edge, Siemens Healthineers, Erlangen, Germany). The scanning parameters included a tube voltage of 80 kV and a fixed effective tube current of 500 mAs. The image acquisition was set at 16 × 0.3 mm with a slice thickness of 0.4 mm, a pitch of 0.4, and rotation time of 1.0 s. Image reconstruction was performed using a sinogram-affirmed iterative reconstruction (SAFIRE, Siemens Healthineers) at level 5 with a sharp edge-enhancing reconstruction kernel (v80U). Obtained DICOM images were analyzed with the ImageJ 1.47v software by obtaining the mean gray values for each slice of the scaffolds. The Phantom EFP-06-96 reference block served as the density reference [38].

### 4.13. Stiffness of the PRP Scaffolds

The scaffold stiffness was determined using a ZwickiLine Z 2.5TN (Zwick GmbH & Co. KG, Ulm, Germany) material testing machine [35]. Scaffolds were compressed repetitively four times uniaxial by 10% (5 mm/min speed) of the original height. A Xforce HP 5N sensor measured the required load in real time. Young’s modulus [MPa] was calculated from the resulting load deformation (stress–strain) curve in the region of linear elastic deformation by dividing the applied force [N] times the initial scaffold height [mm] by the area of the scaffold [mm^2^] times the change in height [mm] [38].

### 4.14. Statistics

The data are presented as box blots with individual data points or line charts, with the median and 95% confidence interval (CI). All experiments were repeated at least three times (biological replicates N ≥ 3) with two to four technical replicates (n = 2–4). All functional data were normalized to the respective total protein content. To minimize variations between the different biological replicates data are displayed as z-scores. The z-scores were calculated according to the formula z = (x − μ)/σ, where ‘x’ is the data point, ‘μ’ is the mean, and ‘σ’ is the standard deviation of the dataset, representing all technical replicates of each biological replicate. Statistical analyses were performed with the GraphPad Prism software version 8.0 (GraphPad Software, La Jolla, CA, USA) using non-parametric assays, as a normal distribution cannot be assumed with the given sample size. *p* < 0.05 was considered statistically significant.

## Figures and Tables

**Figure 1 ijms-26-07699-f001:**
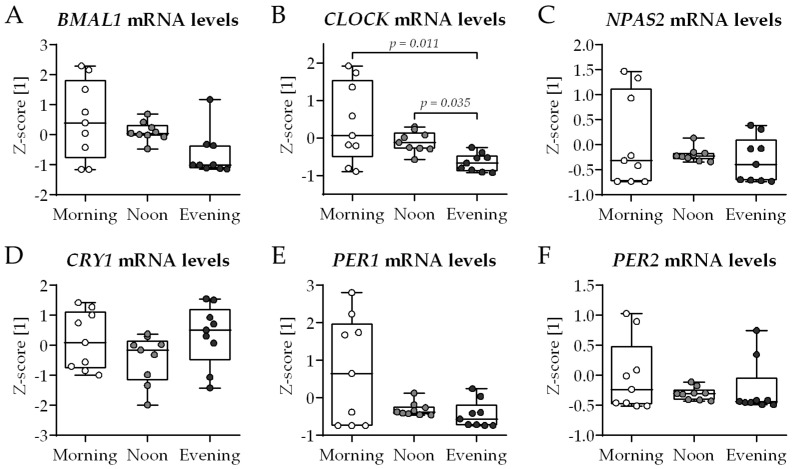
Expression of circadian clock genes in conventional bone cell co-culture is not affected by the time of day. THP-1 and SCP-1 cells were co-cultured in osteogenic differentiation medium containing 2% fetal calf serum. At day 21 of culture, total mRNA was collected in the morning (7–8 am), noon (1–2 pm), or evening (7–8 pm) to investigate expression of genes (qRT-PCR) involved in the core loop of circadian rhythm: (**A**) *BMAL1*, (**B**) *CLOCK*, (**C**) *NPAS2*, and their negative feedback regulators (**D**) *CRY1*, (**E**) *PER1*, and (**F**) *PER2*. Gene expression levels were determined using the ΔΔCt method and normalized as z-score. Data are displayed as box plots with individual data points (N = 4, n = 2) and the median. Groups were compared with the non-parametric Kruskal–Wallis test followed by Dunn’s multiple comparison test when *p* < 0.05. All *p*-values below 0.1 are displayed in the graphics.

**Figure 2 ijms-26-07699-f002:**
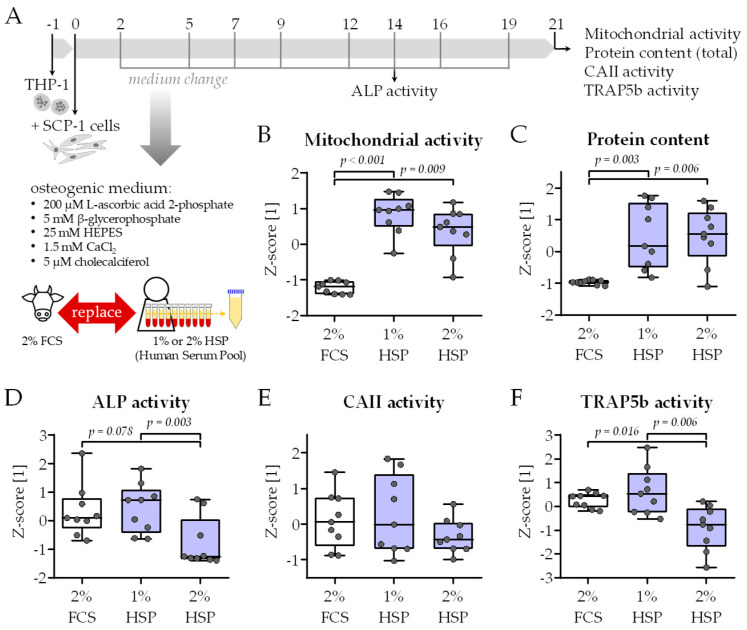
Functional testing of the modified medium composition. (**A**) Schematic overview of the experimental setup. THP-1 and SCP-1 cells were co-cultured in osteogenic differentiation medium containing 2% fetal calf serum (FCS) or 1% respective 2% human serum pool (HSP). For the HSP, equal volumes of human serum from 10 donors (N = 10) were pooled. On day 21 of culture, viability was assessed by (**B**) mitochondrial activity (resazurin conversion assay) and (**C**) total protein content (Sulforhodamin B staining). On day 14 of culture, (**D**) alkaline phosphatase (ALP) activity was measured photometrically as an early osteogenic marker. On day 21 of culture, (**E**) carbonic anhydrase II (CAII) and (**F**) tartrate-resistant acidic phosphatase (TRAP5b) activity were measured photometrically as an early and late osteoclast marker, respectively. All data were normalized as z-scores and are displayed as box plots with individual data points (N = 3, n = 3). Groups were compared with a non-parametric Kruskal–Wallis test followed by Dunn’s multiple comparison test whenever *p* < 0.05. All *p*-values below 0.1 are displayed in the graphics.

**Figure 3 ijms-26-07699-f003:**
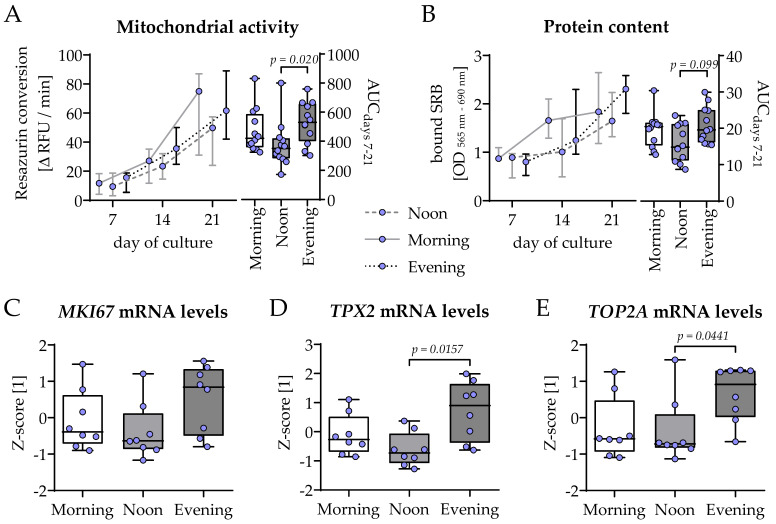
Highest cell proliferation in bone cell co-cultures differentiated in the presence of human serum collected in the evening. THP-1 and SCP-1 cells were co-cultured in osteogenic differentiation medium containing 1% human serum pools (N = 10) obtained in the morning (7–8 am), noon (1–2 pm), or evening (7–8 pm). Cell growth was assessed by changes in (**A**) mitochondrial activity (resazurin conversion assay) and (**B**) total protein content (Sulforhodamin B staining). Changes over time are displayed as line charts (median ± 95% CI of N = 4, n = 3) and summarized as the area under the curve (AUC). On day 21 of culture, total mRNA was collected in the morning and the expression of genes involved in the proliferation was investigated by qRT-PCR (ΔΔCt) and normalized as a z-score: (**C**) *MIK67*, (**D**) *TPX2*, and (**E**) *TOP2A*. Data are displayed as box plots with individual data points (N = 4, n ≥ 2) and the median. Groups were compared with a non-parametric Kruskal–Wallis test followed by Dunn’s multiple comparison test whenever *p* < 0.05. All *p*-values below 0.1 are displayed in the graphics.

**Figure 4 ijms-26-07699-f004:**
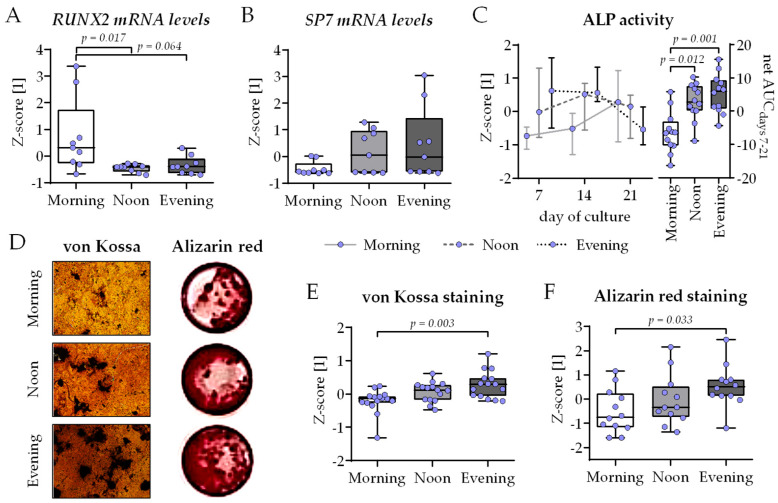
Highest osteoblast function in bone cell co-cultures differentiated in the presence of human serum collected in the evening. THP-1 and SCP-1 cells were co-cultured in osteogenic differentiation medium containing 1% human serum pools (N = 10) obtained in the morning (7–8 am), noon (1–2 pm), or evening (7–8 pm). On day 21 of culture, total mRNA was collected in the morning, and gene expression levels of (**A**) *RUNX2* and (**B**) *SP7* (Osterix) were investigated by qRT-PCR (ΔΔCt). (**C**) Alkaline phosphatase (ALP) activity was measured as an early osteogenic function on days 7, 14, and 21 of culture. Formed mineralized matrix was visualized by (**D**) von Kossa and Alizarin red staining (representative images) and quantified (**E**) by image analysis (ImageJ 1.47v) or (**F**) photometrically, respectively. All data are displayed as box plots with individual data points (N = 4, n ≥ 2 or 3) and the median. Groups were compared with a non-parametric Kruskal–Wallis test followed by Dunn’s multiple comparison test whenever *p* < 0.05. All *p*-values below 0.1 are displayed in the graphics.

**Figure 5 ijms-26-07699-f005:**
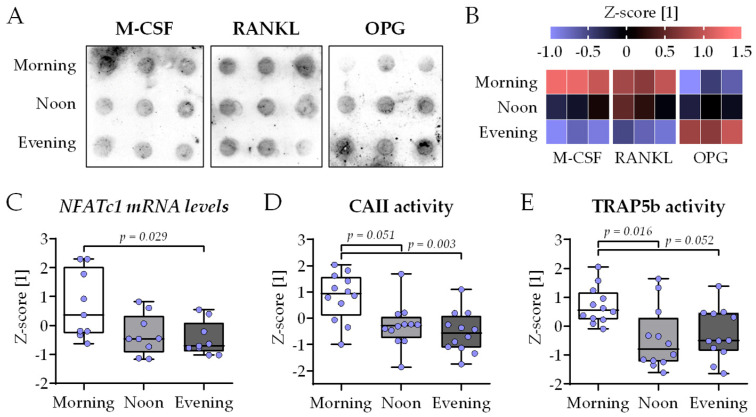
Highest osteoclast function in bone cell co-cultures differentiated in the presence of human serum collected in the morning. THP-1 and SCP-1 cells were co-cultured in osteogenic differentiation medium containing 1% human serum pools (N = 10) obtained in the morning (7–8 am), noon (1–2 pm), or evening (7–8 pm). The levels of the macrophage colony-stimulating factor (M-CSF), receptor activator of NF-κB ligand (RANKL), and Osteoprotegerin (OPG) in the serum pools were determined by dot blot (n = 3). (**A**) Image of signal intensity and (**B**) its quantification by ImageJ 1.47v analysis (heat map). (**C**) On day 21 of culture, total mRNA was collected in the morning, and the gene expression levels of the key osteoclastic transcription factor *NFATc1* were determined by qRT-PCR (ΔΔCt). After 21 days of differentiation, (**D**) carbonic anhydrase II (CAII) and (**E**) tartrate-resistant acidic phosphatase (TRAP5b) activities were measured as early and late osteoclast markers, respectively. All data are displayed as box plots with individual data points (N = 4, n = 2 or 3) and the median. Groups were compared with a non-parametric Kruskal–Wallis test followed by Dunn’s multiple comparison test whenever *p* < 0.05. All *p*-values below 0.1 are displayed in the graphics.

**Figure 6 ijms-26-07699-f006:**
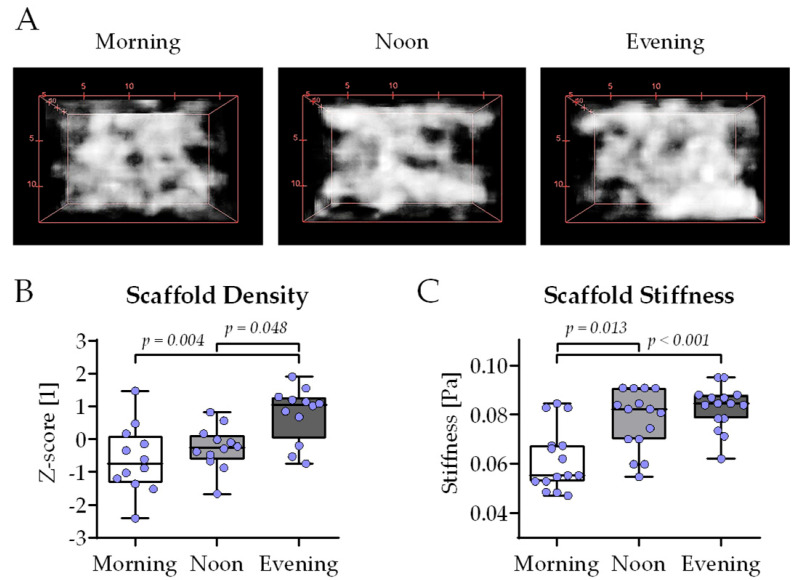
Highest scaffold density and stiffness with bone cell co-cultures differentiated in the presence of human serum collected in the evening. Three-dimensional co-cultures of THP-1 and SCP-1 cells were osteogenically differentiated with media containing 1% human serum pools (N = 10) obtained in the morning (7–8 am), noon (1–2 pm), or evening (7–8 pm). After 21 days of culture, mineral density of the scaffolds was determined by computer tomography. (**A**) Three-dimensional reconstructions of the cultured scaffolds representative for the morning, noon, and evening group. (**B**) The bone mineral density of the cultured scaffolds was quantified using the ImageJ 1.47v software. (**C**) Scaffold stiffness (E-modulus) was determined using a ZwickiLine Z 2.5TN material testing machine. All data were normalized as z-scores and are displayed as box plots with individual data points (N = 4 or 5, n = 3). Groups were compared with a non-parametric Kruskal–Wallis test followed by Dunn’s multiple comparison test whenever *p* < 0.05. All *p*-values below 0.1 are displayed in the graphics.

**Figure 7 ijms-26-07699-f007:**
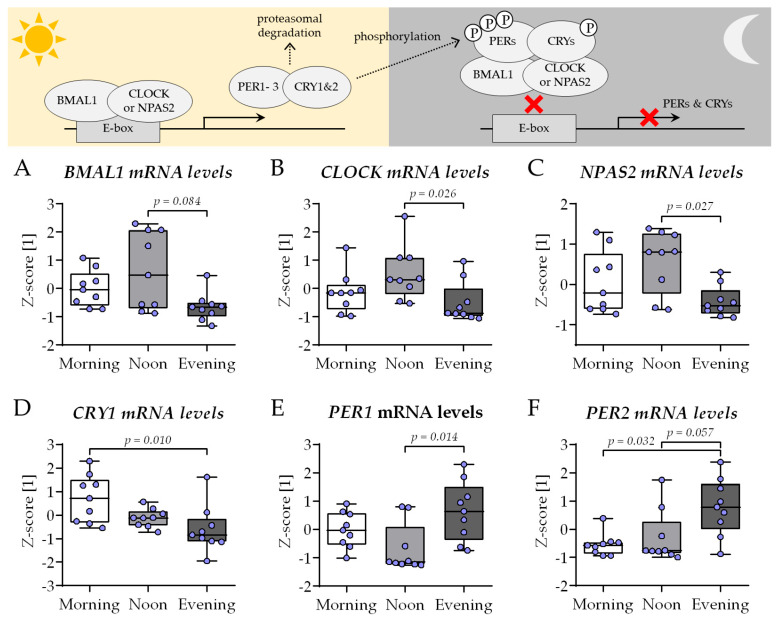
The expression of circadian clock genes in the newly established bone cell co-culture shows changes characteristic for the time of day. THP-1 and SCP-1 cells were co-cultured in osteogenic differentiation medium containing 1% human serum pools (N = 10) obtained in the morning (7–8 am), noon (1–2 pm), or evening (7–8 pm). On day 21 of culture, total mRNA was collected in the morning, and the expression of genes involved in the circadian rhythm core loop (scheme) was investigated by qRT-PCR: (**A**) *BMAL1*, (**B**) *CLOCK*, (**C**) *NPAS2*, and their negative feedback regulators (**D**) *CRY1*, (**E**) *PER1*, and (**F**) *PER2*. Gene expression levels were determined using the delta-delta Ct (ΔΔCt) method and normalized as z-scores. Data are displayed as box plots with individual data points (N = 4, n = 2) and the median. Groups were compared with a non-parametric Kruskal–Wallis test followed by Dunn’s multiple comparison test when *p* < 0.05. All *p*-values below 0.1 are displayed in the graphics.

**Table 1 ijms-26-07699-t001:** List of the primers and quantitative real-time polymerase chain reaction conditions for each gene.

Gene	Role	Accession Number	Forward Primer (5′–3′)	Reverse Primer (3′–5′)	Amplicon Size [bp]	Efficiency [%]	Annealing Temp. [°C]	Cycles [N]
B2M	HKG	NM_004048.2	AGATGAGTAT GCCTGCCGTG	GCGGCATCTT CAAACCTCCA	105	101	60	40
CLOCK	Circadian Rhythm	NM_001267843.2	ACGCACACAT AGGCCATCTT	ATTATGGGTG GTGCCCTGTG	177	102	66	40
BMAL1		NM_001030272.3	TCCTTTGTTG TAGGTGGCCC	GCGATGACCC TCTTATCCTGT	139	112	66	40
NPAS2		NM_002518.4	ACACTCGGTG GTCAGTTACG	CCGATGGCGA ATGACTGGTA	188	110	66	40
CRY1		NM_001413458.1	CCCAGGTTGT AGCAGCAGTG	AGGACGTTTC CCACCACTTG	111	135	60	40
PER1		NM_002616.3	GGGGACCAAG AAAGATCCGC	GCTACACTGA CTGGTGACGG	145	97	64	40
PER2		NM_022817.3	CATCGACGTG GCAGAATGTG	ACGTCTGCTC TTCGATCCTG	161	80	60	40
MKI67	Proliferation	NM_002417.5	CGTCCCAGTG GAAGAGTTGT	CGACCCCGCT CCTTTTGATA	143	102	63	40
TPX2		NM_012112.5	GGAAGCACCA GCTGGAAGA	GAACTAGAGA ACCAGAAAGGCCC	147	110	63	40
TOP2A		NM_001067.4	GTTCTTGAGC CCCTTCACGA	ACCCACATTT GCTGGGTCA	216	123	63	40
RUNX2	Transcription Factor	NM_001024630.4	CTGTGGTTAC TGTCATGGCG	GGGAGGATTT GTGAAGACGGT	170	118	60	40
SP7		NM_152860.1	CCCAGGCAAC ACTCCTACTC	GGCTGGATTA AGGGGAGCAAA	175	91	62	40
NFATC1		NM_172390.2	TGCAAGCCGA ATTCTCTGGT	CTTTACGGCG ACGTCGTTTC	228	87	64	40

## Data Availability

Data is contained within the article. The original contributions presented in this study are included in the article. Further inquiries can be directed to the corresponding author.

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
