# Peer review of "In Vitro Modeling of Diurnal Changes in Bone Metabolism"

_ijms, 2025, doi:10.3390/ijms26167699_

Round 1
Reviewer 1 Report
Comments and Suggestions for Authors
This manuscript from Ehnert and colleagues develops a novel in vitro model to investigate circadian effects on bone physiology. As discussed below I found this to be a very well-done study and identified no concerns around its publication.
Main findings of the study: This model experimental system was based on co-culture of SCP-1 and THP-1 cells, representing osteoblast and osteoclast progenitors. They began by confirming that there are minimal periodic changes in major circadian clock genes. They then sought to validate their experimental system with a change in culture media from 2% fetal calf serum to either 1% or 2% pooled human serum (HSP, serum pooled from multiple donors), finding that 1% HSP gave similar levels of osteoblast and osteoclast marker gene expression as 2% FCS. They then performed cultures with HSP collected either in the morning, midday or in the evening, and measured expression of osteoblast and osteoblast markers, finding highest osteoblast differentiation marker expression in the evening HSP cultures, while osteoclast differentiation markers were increased in the morning HSP cultures. Similarly, 3D cultures with the evening HSP showed the highest scaffold density and stiffness. Finally, the authors measured expression of several circadian clock genes in their system, finding changes that seem to correspond with the HSP timing.
This project represents a clever experimental system seeking to mirror aspects of in vivo circadian rhythm in a culture system. The authors did a good job carefully validating their system before starting their main experiments. The introduction is informative, the results are clearly presented and the discussion of their conclusions is lengthy and well-done. Materials & methods are thorough and detailed. No meaningful deficiencies were noted by this reviewer.
Author Response
We would like to thank the reviewer for carefully reviewing the manuscript and for highlighting the quality of the experiments and the relevance of the findings.
Reviewer 2 Report
Comments and Suggestions for Authors
The manuscript is well-written and logically structured. The introduction provides a comprehensive overview of the field and clearly outlines the need to investigate the effects of circadian rhythms on bone metabolism using an in vitro approach. The results are clearly presented and thoughtfully discussed. The accompanying figures are generally of good quality and are appropriately described. The methods are thoroughly and transparently reported.
Nevertheless, the manuscript has several limitations that should be addressed prior to publication. These include the need to improve the quality of certain figures and to more clearly discuss the translational potential of the model in a clinical context
Specifically,
Figures 2, 3, 5, and 6: The capitalization of parameter labels should be made consistent across all figures. All terms should begin with a capital letter. For example, in Figure 6, both "scaffold" and "Scaffold" are used—this should be standardized throughout.
Figure 4: The sequence of subfigure labels appears inconsistent and illogical. For instance, the order in this figure is A, C, D, B, E, F, whereas in all other figures, the sequence follows the standard alphabetical order (A, B, C, D, E, F). This should be corrected for clarity and consistency.
Figures 4D and 6A: These panels are too small and blurry. They should be improved to ensure adequate resolution and readability before publication.
Discussion section: The limitations regarding the translational relevance of the in vitro findings to in vivo or clinical contexts should be elaborated in more detail.
Reviewer 3 Report
Comments and Suggestions for Authors
- In the study, serum pools (HSPs) of 10 healthy volunteers were used, but baseline information such as the age, gender ratio, and daily routine of the volunteers was not clearly stated. The serum components of different populations may vary, which can affect the stability and repeatability of the model. It is recommended to supplement the detailed baseline data of the volunteers to prove the representativeness of the serum pool.
- Regarding the 3D Co-Cultures in Platelet-Rich Plasma (PRP) Cryogel Scaffolds mentioned in the article, the author should provide a description and clarify whether they can well simulate the osteogenic microenvironment.
- The discussion should include an addition about the practical utilities of how this new model can simulate the physiological environment, such as the recapitulation of the characteristics of a specific circadian rhythm disorder, so as to achieve further expansion.
- The description of data standardization is ambiguous. In several charts (such as Figure 1, Figure 2, Figure 4, etc.), "z-score" and "2-score" are mentioned as data standardization methods, but the specific calculation logic (such as whether it is based on the mean of the control group, the range of sample size, etc.) is not clearly stated in the method part of the legend or the body.
- Do physical properties such as pore size and porosity of PRP cryogel scaffolds influence the osteogenic/osteoclastic functions of cells, thereby interfering with mineral deposition? Is it necessary to supplement the scaffold characterization data (e.g., scanning electron microscopy images)?
- In Figure 6, only the differences between the HSPs treatment groups in the morning, at noon, and in the evening were compared. No "FCS control group" (i.e., the 3D scaffold for traditional 2% FCS culture) was set up, which cannot visually reflect the advantages of HSPs over FCS in 3D models. Why wasn't this control group included? Are there significant differences in mineral density and stiffness between it and the HSPs group?
